# Intracoronary Application of Super-Saturated Oxygen to Reduce Infarct Size Following Myocardial Infarction

**DOI:** 10.3390/jcm11061509

**Published:** 2022-03-09

**Authors:** Andreas Schäfer, Muharrem Akin, Johanna Diekmann, Tobias König

**Affiliations:** 1Department of Cardiology and Angiology, Hannover Medical School, Carl-Neuberg-Str. 1, 30625 Hannover, Germany; akin.muharrem@mh-hannover.de (M.A.); koenig.tobias@mh-hannover.de (T.K.); 2Department of Nuclear Medicine, Hannover Medical School, Carl-Neuberg-Str. 1, 30625 Hannover, Germany; diekmann.johanna@mh-hannover.de

**Keywords:** super-saturated oxygen, acute myocardial infarction, primary PCI, infarct size, secondary prevention

## Abstract

Optimal medical therapy for secondary prevention following acute myocardial infarction reduces non-fatal ischaemic events. Intensive antithrombotic or lipid-lowering approaches have failed to significantly lower mortality. In the past, reduction of infarct size in patients undergoing primary percutaneous revascularisation for acute myocardial infarction had been considered as a surrogate outcome marker. However, infarct size measured by magnetic resonance imaging or SPECT is strongly associated with all-cause mortality and hospitalization for heart failure within the first year after an acute myocardial infarction. Intracoronary administration of super-saturated oxygen (SSO_2_) immediately after revascularisation is an approach that can be used to reduce infarct size and, therefore, improve cardiovascular outcome in patients with acute myocardial infarction. In this article, we describe the modulation of pathophysiology by SSO_2_, review the existing trial data and present our first impressions with the technique in real clinical practice.

## 1. Introduction

Acute myocardial infarction (MI) is one of the major contributors to cardiovascular morbidity and mortality. In the acute phase of ST-elevation MI (STEMI), ventricular arrhythmias and cardiogenic shock are the main drivers of mortality contributing to in-hospital mortality of approximately 10% and time from first medical contact to primary percutaneous coronary intervention (PCI) is a strong predictor of an adverse outcome [1]. However, when surviving the initial pre-hospital and in-hospital phase, annual mortality following STEMI is at a range of 1%/year and remains unaltered by intense lipid-lowering, anti-thrombotic or anti-inflammatory therapies [2,3,4,5,6,7,8].

A meta-analysis of 10 randomized primary PCI trials including more than 2500 STEMI patients showed that infarct size measured within 1 month by cardiac magnetic resonance (CMR) imaging or technetium-99m sestamibi single-photon emission computed tomography (SPECT) was strongly associated with all-cause mortality and hospitalization for heart failure within 1 year [9]. This finding resulted in infarct size being accepted as a surrogate parameter of adverse clinical outcome. Hence, several interventions were recently described aiming for reduction in infarct size following acute STEMI, because animal data suggest that almost half of the infarct size might be attributable to reperfusion injury [10,11]. A central contributor to reperfusion injury seems to be mitochondrial dysfunction [12,13,14,15].

One of the approaches to reduce reperfusion injury was infusion of the mitochondria-targeted peptide, Bendavia, to reduce mitochondria-produced reactive oxygen species, which are believed to be centrally involved in reperfusion injury. While this approach seemed to reduce infarct size in animal models [16], no effect was observed in the human pilot trial [17]. Hypothermia of 32 °C body temperature requiring 18 min delay for primary PCI has also been investigated and provided differing results [18,19]. As the topic of hypothermia is well validated to reduce cerebral reperfusion injury presumably by reducing mitochondria-related oxygen radical formation, it is discussed in more detail in a separate review in this edition [20]. Another approach to target mitochondrial radical formation is left-ventricular unloading using an Impella microaxial pump [21]. This approach requiring 30 min delay for primary PCI has been investigated in an encouraging human pilot trial [22], is currently tested in a larger clinical trial [23] and will be described in more detail in a separate review in this edition. A short summary of therapies previously tested for infarct size reduction with positive preclinical but negative clinical data is provided in Table 1. More in-depth reviews with detailed information on those therapies are available [24,25,26].

## 2. A Brief History of Hyperbaric Oxygen Therapy

As local hypoxia was considered as a driving force for free-radical generation and mitochondrial dysfunction, the first experimental step to counteract this was hyperbaric oxygen therapy (HBO). In the 1950s, initial experiments of coronary sinus occlusion leading to higher coronary sinus pressure resulted in reduced mortality in animal models of coronary arterial occlusion [27]. Hence, the investigators believed that by increasing oxygen content of any blood, plasma or fluid still reaching an ischaemic zone, they would reduce ischaemic burden. They postulated that already a small increase in oxygen by one volume-% at double the atmospheric pressure would result in a fifteen-fold increment of the oxygen in solution and then provide almost normal oxygen supply in the ischaemic zone [28]. In 1964, Chardrack and colleagues performed coronary ligation procedures in 162 dogs divided into four experimental groups to assess whether 100% oxygen by ventilation at 2 or 4 atmospheres of pressure in a hyperbaric chamber would reduce infarct size and improve survival. They observed an increase of survival from 52.5% on room air at 1 atmosphere to 77.8% on 100% O_2_ at 4 atmospheres [29]. Similarly, Peter et al. showed that moderate but prolonged HBO could lengthen life during and after coronary occlusion in pigs [30]. These experimental data encouraged physicians in the 1960s to start treating acute MI patients with HBO [31]. In 1973, Thurston and colleagues reported a randomized controlled clinical trial on HBO, in which 103 patients with acute MI received HBO with an intention of minimum 4 h compared to 105 controls. Following adjustment for the distribution of severely ill patients between both groups (patients with high Peel index profited more), they observed a significant mortality reduction from 22.9 to 16.5% [32].

In 1993 Sterling et al. started to investigate in a rabbit model whether HBO affects infarct size. They surgically occluded one branch of the left coronary artery for 30 min prior to reperfusion. In their experimental setup they observed significant infarct size reduction from 42 to 10% when both ischaemia as well as reperfusion occurred during hyperbaric conditions and to 15% when only reperfusion was hyperbaric [33]. At the same time, Sirsjö et al. observed that HBO enhanced recovery of blood flow in postischemic muscle tissue, indicating attenuation of microvascular dysfunction or damage in the postischemic period [34]. In 1994, Cason and colleagues investigated whether therapeutic hyperoxia diminishes myocardial stunning by cannulating the left anterior descending coronary artery and perfusing it using an extracorporeal circuit. Myocardial stunning was created by reducing coronary artery flow to 40% of control values for 30 min. Hyperoxia was used by replacing nitrogen with oxygen and increasing oxygen partial pressure to 380 mmHg. They concluded that hyperoxia used as a treatment during transient myocardial ischemia diminished postischaemic myocardial stunning [35].

## 3. From Hyperbaric Oxygen Therapy to Aqueous Oxygen Infusion

Placing patients with acute STEMI into HBO chambers might pose several logistical problems including poor accessibility, difficulties for monitoring, claustrophobia and necessary decompression phases when performing the treatment for multiple hours. Blood hyperoxygenation might be possible using conventional extracorporeal membrane oxygenation (ECMO). However, this approach is limited by invasiveness, activation of coagulation and inflammation and the need for specialized personal for monitoring. It would counteract the minimal-invasive approach using radial access for prevention of bleeding when performing primary PCI nowadays. A solution for all those problems would enable easy hyper-oxygenation of blood components able to reach the peripheral myocardium. Aqueous oxygen has an oxygen-carrying capacity which is approximately 10-fold greater than those of blood. The advantage of aqueous oxygen is that it can be infused into other aqueous media at normal pressure without formation of gas bubbles [36]. Following technical evolution, it is possible to inject a stabilized aqueous oxygen solution on a saline basis into blood, which is drawn from a patient’s peripheral artery and afterwards recirculated into a coronary artery. The subsequent hyperoxemic perfusion is able to reach regions with impaired erythrocyte flow due to microvascular obstruction and reverse persistent tissue ischaemia. It is assumed that hyperoxemic perfusion will stop typical hypoxia-related changes by preventing cell ischaemia and cell edema [36].

The TherOx^®^ system (TherOx Inc, Irvine, CA, USA) produces aqueous oxygen from bottled oxygen for medical use and sterile sodium-chloride solution. It requires arterial blood either from a singular femoral 4F-sheath in case of trans-radial catheterization or escalation of femoral access to a 7F-sheath which can host both arterial drainage as well as the guiding 5F-catheter for coronary reinfusion of hyperoxaemic blood. For easy handling in the cath-lab, a single-use sterile cartridge is placed in the TherOx^®^ system and connected with a saline infusion as well as one line for drawing arterial normoxic blood and one line to connect with the coronary guiding catheter placed in the left-main coronary artery. Within the cartridge, aqueous oxygen solution was produced and injected into the circulating blood. The guiding catheter then delivers hyperoxaemic blood to the coronary system [36].

The proof of principal for aqueous oxygen therapy was provided by Spears et al. in an animal model using dogs subjected to coronary hypoxaemia. In that model, intra-arterial aqueous oxygen preserved left-ventricular ejection fraction [37]. In a pig model, the same investigators later demonstrated that intracoronary hyperbaric reperfusion using aqueous oxygen attenuated myocardial ischemia/reperfusion injury [38]. Overall, several preliminary animal studies suggested that aqueous oxygen-derived hyperoxaemic reperfusion was safe, corrected regional hypoxaemia, raised oxygen partial pressures without bubble formation, and improved left-ventricular function with a potential to reduce infarct size [36]. While the issue of additive ambient oxygen supply was controversially discussed in the past based on missing beneficial effects on major adverse cardiovascular events (MACE) [39] and potential increase in myocardial injury and infarct size [40], the concept of providing excessive amounts of oxygen by SSO_2_ appears to be controversial at first sight. However, animal experiments suggest that aqueous oxygen does not result in an increased formation of reactive oxygen species but rather reduces them in an animal setting [38].

Experimental data support the pathophysiological hypothesis that microvascular ischemia caused by microvascular obstruction contributes to myocardial tissue injury over a prolonged time during reperfusion. It is believed that SSO_2_ provides hyperoxemic levels of oxygen, which are up to 10 times higher than normal, of dissolved oxygen when it is infused in the infarct-related artery. This allows for high oxygen diffusion before flow is restored downstream. Increased oxygen delivery via the plasma to endothelial cells can prevent severe cell edema, allow better perfusion and correct persistent myocardial ischemia, particularly in regions where erythrocyte flow is impaired by compromise of the microvascular luminal integrity [41]. As the diffusion through plasma appears to be a central issue for successful prevention of myocardial injury, it also provides a pathophysiological rational why SSO_2_ in contrast to LV unloading or hypothermia either requires or benefits from revascularisation prior to application. On the other hand, this provides the clinical advantage of not delaying primary PCI compared to the other methods [20,22].

## 4. Study Results for Super-Saturated Oxygen Therapy in STEMI

After the encouraging results from animal experiments, first smaller pilot trials were started in humans with acute MI. A recent review extensively describes the evolvement from hypothesis generation over introduction of the SSO_2_ concept and preclinical studies to clinical studies finally leading to FDA approval [41].

### 4.1. Pilot Trials

Initial experience with aqueous oxygen in humans using the TherOx^®^ device was reported in 2002 by an American–Italian multicentre investigation. In this pilot trial, 29 patients with acute MI undergoing primary PCI received hyperoxaemic blood infusion into the infarct-related artery for 60–90 min after successful revascularisation. Hyperoxemic reperfusion was well tolerated by all patients and neither clinical, electrical nor hemodynamic instability was observed during the aqueous oxygen infusion. The primary purpose of the study was to evaluate the feasibility and safety of performing hyperoxemic reperfusion in patients undergoing primary PCI for acute MI. While safety could be demonstrated, it remained unclear whether there was a potential benefit regarding reduction of infarct size or improvement in left-ventricular function [42].

An Italian study reported 27 patients with acute anterior AMI treated with primary PCI within 12 h after symptom onset receiving selective hyperoxemic perfusion into the left anterior descending coronary artery for 90 min. They were compared to 24 anterior STEMI patients matched in clinical and angiographic characteristics and treated with conventional primary PCI. Recanalization and hyperoxemic perfusion were successfully performed in all patients. After hyperoxemic perfusion patients showed shorter time-to-peak creatine kinase release and a higher rate of complete ST-segment resolution. A significant improvement of mean LV ejection fraction from 44 to 55% was observed after 3 months in hyperoxemic perfusion patients only. Therefore, the authors concluded that hyperoxemic perfusion using the aqueous oxygen technique is associated with LV function recovery [43].

### 4.2. Clinical Trials

Table 2 provides an overview of clinical trials using hyperoxemic therapy in AMI.

The first prospective, randomized trial of intracoronary hyperoxemic reperfusion after primary PCI was the Acute Myocardial Infarction With Hyperoxemic Therapy (AMIHOT I) trial published in 2007 [44]. In AMIHOT I, 269 patients with acute anterior or large inferior AMI undergoing primary or rescue PCI within 24 h from symptom onset were randomized to either hyperoxemic reperfusion or normoxemic blood autoreperfusion. Hyperoxemic reperfusion was performed for 90 min using intracoronary aqueous oxygen through an intracoronary infusion catheter positioned in the proximal segment of the infarct-related artery. There was no difference in major adverse cardiac events (MACE, composite of death, reinfarction, target vessel revascularization or stroke). While there was no difference in the change in regional wall motion by echocardiography after 3 months, there was a significant improvement in regional wall motion in patients with anterior AMI treated within the first 6 h after symptom onset, which was accompanied by a significant reduction in infarct size determined after 14 days by technetium-99m sestamibi SPECT [44]. While demonstrating feasibility and safety, this trial identified the potential patient cohort that is most likely to benefit from the treatment: the acute anterior AMI presenting within 6 h after symptom onset.

Based on the positive findings in the anterior AMI group treated within 6 h in AMIHOT I, a second prospective, multicentre trial (AMIHOT II) was conducted only randomizing patients with anterior AMI and symptom onset < 6 h prior to primary PCI and reported in 2009. Of 301 patients randomized in the intention-to-treat analysis, 222 received SSO_2_ and 79 were controls. The analysis was planned to include all patients from both groups in AMIHOT I meeting the AMIHOT II inclusion criteria for a pooled analysis, which finally included 124 control and 258 SSO_2_ therapy patients. There was no significant difference in MACE rate, neither in AMIHOT II alone nor in the pooled group. Regarding non-MACE adverse events, the rate of hemorrhagic adverse events was higher with SSO_2_ than in controls (24.8% vs. 12.7%, *p* = 0.03) predominantly attributable to access site related bleeding. However, at that time 9F sheaths were used in 53%, 8F sheaths in 22%, and 7F sheaths in 25% of patients. During the trial, the type of catheter used for SSO_2_ therapy did change accompanied by preferring a single unilateral instead of dual bilateral femoral artery access. This resulted in less access site bleeding complications on SSO_2_ during the later phase of the trial compared to the initial phase. On the ischemic site of events, the rate of stent thrombosis trended to be higher in the SSO_2_ group (4.1% vs. 2.5%, *p* = 0.73), but it has to be acknowledged that in AMIHOT II patients received early-generation drug-eluting stents (numerically more in the SSO_2_ group, 52% vs. 56%) and the P2Y12 inhibitor used had been clopidogrel [45]. In AMIHOT II, SSO_2_ therapy reduced infarct size determined after 14 days by ^99m^Tc-sestamibi SPECT from 26.5–20.0% (adjusted *p* = 0.03). In the pre-planned pooled analysis form both AMIHOT-trials, infarct size was reduced from 25.0–18.5% (*p* = 0.02). Among 154 patients with a baseline LV-EF below 40%, SSO_2_ therapy reduced infarct size from 33.5–23.5%. Therefore, the overall conclusion from both AMIHOT trials was that intracoronary SSO_2_ infusion into the infarct-related artery following successful primary PCI is feasible and safe and reduces myocardial infarct size in patients with acute anterior AMI presenting within 6 h of symptom onset [45]. In addition to identifying the early presenting anterior AMI as best candidate for SSO_2_ therapy, this trial indicates that patients with apparent ventricular compromise (LV-EF < 40%) suggestive for larger infarct zone are the ones profiting most from the intervention.

To overcome the limitation of selective infusion catheters placed within the proximal segment of the infarct-related coronary artery, the idea of applying SSO_2_ directly via a guiding catheter in the left-main coronary artery evolved. Therefore, the IntraCoronary Hyperoxemic Oxygen Therapy (IC-HOT) trials was conducted. IC-HOT, published in 2019, was a single-arm, safety trial including 100 patients with anterior STEMI presenting within 6 h of symptom onset [46]. The primary endpoint was the 30-day composite rate of net adverse clinical events (NACE; death, reinfarction, clinically driven target vessel revascularization, stent thrombosis, severe heart failure or TIMI major/minor bleeding). Cardiac magnetic resonance imaging was performed at 4 and 30 days to assess infarct size. SSO_2_ delivery was successful in 98% of patients. NACE rate at 30 days was 7.1%. Infarct size was 24% at 4 days and 19% at 30 days. Therefore, the conclusion was that infusion of SSO_2_ via the left-main coronary artery was feasible and safe [46].

The clinical findings observed in the three studies are summarized in Table 3.

### 4.3. Experience in Everyday Clinical Practice

At Hannover Medical School, in March 2021 we had the opportunity to be the first in Europe to gain experience with the commercially available TherOx^®^ system for application of SSO_2_ therapy in everyday clinical routine. In terms of practicability during acute STEMI treatment in the cath-lab, the SSO_2_ technique has one major advantage over other approaches aiming for infarct size reduction in large anterior MIs such as LV-unloading by Impella [22] or therapeutic hypothermia [19]: the additive application is only started after successful revascularization, so it does not delay primary PCI. In the beginning, we had a core team to perform the SSO_2_ therapy. While the assigned operator performed routine primary PCI, the core team set up the TherOx^®^ system. After initial training, it was easily possible to exchange the femoral sheath in case of primary femoral access or to place the additional femoral 4F-sheath in case of a primary radial approach. The perfusion catheter, which is a 5Fr coronary diagnostic catheter, is rapidly placed in the left main coronary artery, while the TherOx^®^ system is primed with blood, and SSO_2_ therapy can start within 4 min after the last angiography at the end of primary PCI. The patients tolerated the 60 min of SSO_2_ administration well. The downside in a busy cath-lab might be that the patients stay on the cath-lab table for one hour of SSO_2_ therapy.

To evaluate further the potentially beneficial effect of SSO_2_ in anterior AMI patients, all our patients undergo distinct imaging for LV function and detection of myocardial inflammation and necrosis using transthoracic echocardiography, cardiac magnetic resonance imaging and ^99m^Tc- tetrofosmin perfusion SPECT [47,48]. Once enough patients have been treated, we plan to analyse the data and perform matched-pair analyses with historic controls without SSO_2_ therapy. In the meantime, German authorities have demanded that prior to reimbursement a large randomized controlled trial with patient-relevant outcome measures has to be conducted to prove the additive benefit suggested by the previous smaller clinical trials.

Overall, SSO_2_ therapy is easy to set up and applicable during standard clinical treatment of acute AMI patients by primary PCI. The application does not delay standard procedures prior to revascularization and can start within minutes after PCI.

## 5. Discussion

SSO_2_ therapy using aqueous oxygen demonstrated proof of principle in animals and showed feasibility and safety in humans undergoing primary PCI for treatment of acute anterior STEMI. While larger clinical trials powered to demonstrate reduction of major adverse cardiovascular events directly are still lacking, evidence from almost 360 patients indicates a potential to reduce infarct size at least in anterior STEMI receiving treatment within 6 h of symptom onset. Aggregate data from AMIHOT I, AMIHOT II, and IC-HOT suggest that SSO_2_ therapy might reduce infarct size in those patients from approximately 25 to 27% of the LV to about 19% of the LV [44,45,46]. Nowadays, reduction in infarct size is a surrogate for all-cause mortality and hospitalization for heart failure within 1 year following acute STEMI. Based on aggregate data from 10 contemporary randomized primary PCI trials, every 5% increase in infarct size was independently associated with a 19% increase in 1-year all-cause mortality and a 20% increase in 1-year hospitalization for heart failure [9]. Therefore, the reduction in infarct size by approximately absolute 5% of LV, as observed under SSO_2_ therapy, would reflect an approximate 17% reduction in all-cause mortality and heart failure hospitalization within 1 year. Considering 1-year mortality or heart failure hospitalization rates in these patients of 2–3% each, an adequately powered randomized controlled outcome trial would have to be unfeasibly large and, therefore, difficult or even impossible to conduct. Using infarct size in early presenting anterior STEMI based on the effects of the older trials, an adequately powered, randomized-controlled trial is feasible with approximately 200 patients per group.

It will be important to identify patients who are ideally suited to profit from SSO_2_ therapy. The earlier trials showed some development from trial to trial. AMIHOT I identified the anterior MI presenting within 6 h of symptom onset as the subgroup profiting most from the procedure [44]. AMIHOT II identified patients with impaired LV function (LV-EF < 40%) indicating a large anterior infarct as a subgroup with more than average benefit. Patients with LV-EF < 40% not receiving SSO_2_ therapy had larger infarct size (33.5% vs. 16.5% of LV) than those with preserved LV-EF and SSO2 reduced infarct size by an absolute 10% [45]. Identifying patients with an absolute 10% infarct size reduction by SSO_2_ could translate into a clinical benefit of almost 36% regarding reduction of mortality and heart failure hospitalization within 1 year [9]. Finally, IC-HOT confirmed feasibility and safety of SSO_2_ therapy while avoiding selective catheterization of the left-anterior descending coronary artery by simply reperfusing hyperoxemic blood through a guiding catheter into the left-main coronary artery, making the procedure extremely easy and rapid to conduct [46].

In comparison to other approaches aiming for infarct size reduction in AMI, which almost all focus on large anterior STEMI within 6 h of symptom onset, SSO_2_ therapy is the least invasive one. Using Impella devices for LV unloading starting prior to and delaying revascularization requires a femoral 14F access [22]. Performing systemic hypothermia of 33 °C also starts prior to and delays reperfusion and is rather uncomfortable for the patient requiring additive medication to tolerate the procedure [19]. Furthermore, hypothermia affects coagulation and potentially contributes to platelet activation. More recently, the CAMI-1 pilot trial, in which C-reactive protein apheresis was conducted within 36 h of symptom onset, failed to demonstrate infarct size reduction or LV function improvement in 45 apheresis-treated compared to 38 control patients. An advantage of the apheresis setting is that it does not delay revascularization and only starts on ICU; however, the delay might have contributed to the lack of effect [49].

In the future, it might be interesting to see whether a combination of different approaches reducing infarct size in addition to primary PCI will provide additive benefit to the patient. However, this will require that the individual interventions provide a benefit on their own. Respective clinical trials are either on their way or in the stage of planning. Theoretically, it is appealing to think about combining these strategies, for example, by infusing SSO_2_ through cooled blood providing additive intracoronary hypothermia [20] or applying it while the LV is unloaded by a microaxilar flow-pump that reduces microvascular resistance and might thereby improve plasma flow and oxygen diffusion [50].

## 6. Conclusions

In summary, intracoronary application of SSO_2_ therapy in acute STEMI is feasible and safe. It does not delay state-of-the art revascularization and can start within 5 min after the end of primary PCI. Earlier studies demonstrated an approximate absolute 6–7% reduction of infarct size in anterior STEMI, which translates into almost 20% reduction of 1-year all-cause mortality and heart failure hospitalization. However, an adequately powered clinical trial under contemporary PCI conditions is needed.

## Figures and Tables

**Table 1 jcm-11-01509-t001:** Previous therapeutic strategies to reduce infarct size, which were successful in experimental models but failed in clinical trials.

Principle to Reduce Infarct Size	Therapies Tested
Endogenours Cardioprotection	Several studies on post- or remote ischemic conditioning, small study on cyclosporine, negative multicentre study for PKC inhibitor, negative trials for intracoronary adenosine.
Reduction of coronary thrombus burden	Mechanical removal of thrombus by routine aspiration, small-molecule glycoprotein IIb/IIIa inhibitors improving coronary flow with varying data on infarct size. Abciximab reducing transmurality of infarction, potentially reducing infarct size compared to thrombectomy or eptifibatide, less new-onset heart failure.
Mechanical protection	Intra-aortic balloon pump without benefit on infarct size.
Slowing metabolism	Intravenous and endovascular cooling indicating a potential in small series, but rapid cooling prior to reperfusion yet not successful in larger trials.
Pharmacological	Agents tested include erythropoietin, glucose-insulin-potassium infusion, statins, several classes of antidiabetic drugs, fibrin-derived peptide, iron chelation, ranolazine, and mitochondria-targeted peptides.

PKC = protein kinase C.

**Table 2 jcm-11-01509-t002:** Clinical trials using hyperoxemic therapy in acute myocardial infarction.

	AMIHOT I	AMIHOT II	IC-HOT
Year of publication	2007	2009	2019
Patients (n)	269	301	100
Location of MI	Anterior or large inferior	Anterior	Anterior
Symptom onset	≤24 h	≤6 h	≤6 h
Reperfusion coronary artery	Proximal infarct- related artery	Proximal LAD	Left-main
Duration of SSO_2_	90 min (87%), 60–89 min (4%), <60 min (8%)	≥90 min (85%), 60–89 min (4%), <60 min (11%)	≥60 min (94%), <60 min (6%)

LAD = Left anterior descending.

**Table 3 jcm-11-01509-t003:** Clinical findings from three randomized trials using SSO_2_.

	AMIHOT I		AMIHOT II		AMIHOT I+II		IC-HOT
	Control	SSO_2_	Control	SSO_2_	Control	SSO_2_	SSO_2_
Patients (n)	135	134	79	222	124	258	100
30-day MACE	7 (5.2%)	9 (6.7%)	3 (3.8%) *	12 (5.4%) *	-	-	1 (1.0%)
Infarct size for anterior AMI	23.0 (5.0; 37.0)	9.0 (0; 30.0)	26.5 (8.5; 44.0)	20.0 (6.0; 37.0)	25.0 (7.0; 42.0)	18.5 (3.5; 34.5)	19.4 (8.8; 28.9)
Infarct size for LV-EF < 40%	30 ± 26	20 ± 30	33.5 (17.5; 38.5)	23.5 (7.5; 38.5)	-	-	-

* in the per-protocol population MACE rate was 3.8% in both groups; Infarct sizes are reported as mean ± SD or median (interquartile ranges) as originally reported. Infarct sizes were determined by ^99m^Tc-sestamibi SPECT in AMIHOT studies and by cardiac magnetic resonance imaging in IC-HOT.

## Data Availability

Not applicable.

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
