# Peer review of "Intracoronary Application of Super-Saturated Oxygen to Reduce Infarct Size Following Myocardial Infarction"

_jcm, 2022, doi:10.3390/jcm11061509_

Round 1

Reviewer 1 Report

We great interest I read the muanuscript. It a field of research which is very important and may ultimately change the way we treat patient with AMI.

Some minor concerns should be mentioned:

  1. Figure 1 and 2 are somewhat unneccesary. It is an example of MRI and SPECT of patient treated with SSO2 and control. Off course there are also patients treated with SSO2 with much larger infarction than the patient displayed. Avoid cherry picking of such MRI's. I would delete.
  2. Could you eloborate on applying SSO2 before reperfusion (and reperfusion injury), while the coronary artery is still occluded with an OTWB for example. 
  3. I would discuss future research ideas including combinations with SSO2 more in a separate paragraph.

Author Response

  1. Figure 1 and 2 are somewhat unnecessary. It is an example of MRI and SPECT of patient treated with SSO2 and control. Off course there are also patients treated with SSO2 with much larger infarction than the patient displayed. Avoid cherry picking of such MRI's. I would delete.

As requested by the reviewer, we have deleted both figures of individual patient data.

  1. Could you elaborate on applying SSO2 before reperfusion (and reperfusion injury), while the coronary artery is still occluded with an OTWB for example. 

This question is theoretically appealing, however, we see that delaying reperfusion raises concerns among interventional cardiologist both when we discuss the trial settings for either cooling in AMI or for LV unloading with Impella and delaying revascularisation for further 30 minutes. Regarding SSO2 on a practical point this would mean putting an OTWB to the infarct-related artery prior to reperfusion, which implies a distally occluded artery. This, however, would negatively affect the ability for hyperoxic diffusion through the plasma, which is considered as a central principle by which SSO2 works. Given the assumption that the target readership of the article are clinicians, also when –as the guest editor for this special issue of JCM – I was inviting others to submit I tried to put the focus on techniques that are either clinically available or are tested in clinical trials at the moment. Nevertheless, we added a short statement why we believe pre-revascularisation SSO2 not being particularly helpful (lines 149-154).  

  1. I would discuss future research ideas including combinations with SSO2 more in a separate paragraph.

As suggested, we added a new paragraph on combinations of SSO2 with particularly LV unloading by Impella or intracoronary hypothermia at the end of the discussion section (lines 326-333).

Reviewer 2 Report

In this manuscript, the authors review the basic principles and clinical studies of supersaturated oxygen (SSO2) therapy following primary PCI in patients with acute myocardial infarction. In their introduction, they correctly point out that, despite primary PCI significantly reduces death rate and heart failure, there is still an important unmet need, namely reduction of myocardial infarct size. After a brief evaluation of some previous approaches to reducing reperfusion injury and improving recovery of left ventricle function, they discuss earlier efforts with hyperbaric oxygen therapy that led to the development of the concept of intracoronary infusion of aqueous SSO2 after mechanical reperfusion therapy in patients with STEMI. A short report on their real-world experience with SSO2 is also included.

I have the following comments:

  1. The article is well written and I believe that the topic is of interest to the cardiology and interventional community. However, it should be noted that a similar review paper has been recently published (Kloner et al. J Am Coll Cardiol Basic Trans Science 2021;6:1021-1033). Indeed, Kloner et al. thoroughly describe the preclinical studies on this technology that laid the foundations for the following clinical trials that have been completed up to now demonstrating that SSO2 after primary PCI is not only safe but is also efficacious to reduce infarct size, thus preserving LV function and reducing adverse LV remodeling. I think that this state-of-the art review should be acknowledged.  

  1. The authors in their introduction should present a brief summary (may be in the form of a table) of the different therapies previously tested in order to reduce the infarct area, which have worked in experimental models but have failed once used in patients. The potential reasons for these failures should also be recalled. See the reviews already published on this topic (Gerczuk PZ, Kloner RA J Am Coll Cardiol 2012;59:969-978; Kloner RA Circ Res. 2013;113:451-463; Kloner RA et al. Cardiovasc Drugs Ther 2017;31:53-61).

  1. Reviewing the AMIHOT II trial, the authors should mention that patients treated with SSO2 during the early phase of the study had an increase in access site-related events (mainly hematomas) likely due to a larger infusion catheter and bilateral femoral access. Moreover, a trend towards a higher rate of stent thrombosis (4.1% vs. 2.5% in the control group) was also observed.

  1. I believe that a key factor leading to reduced contractile function in the infarct zone is poor microvascular perfusion even after normalization of coronary flow by angiographic criteria following primary PCI. Although reperfusion injury may be involved, failure to re-establish adequate tissue perfusion is more likely due to ischemia-induced microvascular damage, and plugging of the microcirculation by thrombus and plaque during mechanical recanalization. Experimental data support the hypothesis that reperfusion microvascular ischemia due to microvascular obstruction contributes to myocardial tissue injury over a prolonged time. In this regard, the authors should mention the proposed mechanism of action of SSO2. Indeed, hyperoxemic levels (7 to 10 times higher than normal) of dissolved oxygen infused in the infarct-related artery allow high O2 diffusion before flow is restored downstream. Increased O2 delivery via the plasma to the endothelial cells prevents their severe swelling, allowing better perfusion and correcting persistent myocardial ischemia, particularly in regions where erythrocyte flow is impaired by compromise of the microvascular luminal integrity.

  1. In view of what I mentioned in point 3, I think that the title of the manuscript is not appropriate as SSO2 rather than reducing reperfusion injury has its main action in treating microvascular dysfunction and reperfusion microvascular ischemia.

  1. Regarding reperfusion injury, the infusion of hyperoxemic blood after infarct-related artery reperfusion could theoretically exacerbate the burst of reactive oxygen species, which are known to generate upon reperfusion of previously ischemic myocardium. If this were really what happens, myeloperoxidase levels would be higher with hyperoxemic reperfusion. However, an animal study (Spears et al. J Invasiv Cardiol 2002;14:160-166) demonstrated just the opposite, showing that this therapy actually reduce reactive oxygen radical damage. This is a reassuring finding, consistent with the observed beneficial effect of SSO2 on infarct size, and should be mentioned in the manuscript.

  1. The clinical experience of the authors with the SSO2 therapy is only reported with a rough description (feasibility with no delay of primary PCI, no MACE during follow-up and a mention only regarding infarct size rduction). Indeed, no detailed data are presented (number of patients, patient characteristics, enrolling criteria and end-points, infarct location, imaging (CMR, rest perfusion SPECT9) data, periprocedural complications, infarct size reduction over time and clinical outcome), limiting the scientific value of this experience. Because the clinical experience with “optimized” SSO2 delivery (i.e. hyperoxemic blood infused through a 5Fr coronary diagnostic catheter into the ostium of the LM, infusion time reduced from 90 to 60 minutes increasing the flow rate from 75 mL/min to 100 mL/min) is limited (20 patients in the pilot study, 100 patients in the IC-HOT study), additional clinical data are needed to confirm the safety and salutary benefits of the “optimized” SSO2 delivery following primary PCI in patients with anterior STEMI.

Author Response

In this manuscript, the authors review the basic principles and clinical studies of supersaturated oxygen (SSO2) therapy following primary PCI in patients with acute myocardial infarction. In their introduction, they correctly point out that, despite primary PCI significantly reduces death rate and heart failure, there is still an important unmet need, namely reduction of myocardial infarct size. After a brief evaluation of some previous approaches to reducing reperfusion injury and improving recovery of left ventricle function, they discuss earlier efforts with hyperbaric oxygen therapy that led to the development of the concept of intracoronary infusion of aqueous SSO2 after mechanical reperfusion therapy in patients with STEMI. A short report on their real-world experience with SSO2 is also included.

We want to thank the reviewer for his/her knowledgeable review.  

  1. The article is well written and I believe that the topic is of interest to the cardiology and interventional community. However, it should be noted that a similar review paper has been recently published (Kloner et al. J Am Coll Cardiol Basic Trans Science 2021;6:1021-1033). Indeed, Kloner et al. thoroughly describe the preclinical studies on this technology that laid the foundations for the following clinical trials that have been completed up to now demonstrating that SSO2after primary PCI is not only safe but is also efficacious to reduce infarct size, thus preserving LV function and reducing adverse LV remodeling. I think that this state-of-the art review should be acknowledged.  

We agree with the reviewer that the recent review by Kloner et al provides a good in-depth description from hypothesis generation, concept introduction, preclinical and clinical studies all the way to final FDA approval. As suggested, we do now cite this article prior to the discussion of the study results (section 4, line 141-149).

  1. The authors in their introduction should present a brief summary (may be in the form of a table) of the different therapies previously tested in order to reduce the infarct area, which have worked in experimental models but have failed once used in patients. The potential reasons for these failures should also be recalled. See the reviews already published on this topic (Gerczuk PZ, Kloner RA J Am Coll Cardiol 2012;59:969-978; Kloner RA Circ Res. 2013;113:451-463; Kloner RA et al. Cardiovasc Drugs Ther 2017;31:53-61).

The review articles suggested by the reviewer are an extensive compilation for those interested in details of those previously failed concepts. We refer to them as suggested (lines 55-57) and tried to extract the essentials without shifting away too much from the main topic of our article by including the suggested table (New Table 1, line 60).

  1. Reviewing the AMIHOT II trial, the authors should mention that patients treated with SSO2during the early phase of the study had an increase in access site-related events (mainly hematomas) likely due to a larger infusion catheter and bilateral femoral access. Moreover, a trend towards a higher rate of stent thrombosis (4.1% vs. 2.5% in the control group) was also observed.

We thank the reviewer for this important addendum, which we added to the AMIHOT II discussion. The issue of stent thrombosis has now also been added including a referral to the type of DES and P2Y12 inhibitor used at the time the trial had been conducted (lines 214-221).

  1. I believe that a key factor leading to reduced contractile function in the infarct zone is poor microvascular perfusion even after normalization of coronary flow by angiographic criteria following primary PCI. Although reperfusion injury may be involved, failure to re-establish adequate tissue perfusion is more likely due to ischemia-induced microvascular damage, and plugging of the microcirculation by thrombus and plaque during mechanical recanalization. Experimental data support the hypothesis that reperfusion microvascular ischemia due to microvascular obstruction contributes to myocardial tissue injury over a prolonged time. In this regard, the authors should mention the proposed mechanism of action of SSO2. Indeed, hyperoxemic levels (7 to 10 times higher than normal) of dissolved oxygen infused in the infarct-related artery allow high Odiffusion before flow is restored downstream. Increased O2delivery via the plasma to the endothelial cells prevents their severe swelling, allowing better perfusion and correcting persistent myocardial ischemia, particularly in regions where erythrocyte flow is impaired by compromise of the microvascular luminal integrity.

We thank the reviewer for these very valuable suggestions and have added them at the end of chapter 3 (lines 141-149).

  1. In view of what I mentioned in point 3, I think that the title of the manuscript is not appropriate as SSO2rather than reducing reperfusion injury has its main action in treating microvascular dysfunction and reperfusion microvascular ischemia.

We acknowledge the reviewer’s point. In order to keep the title clinically focused we have rephrased the title now mentioning the intention to “reduce infarct size” following myocardial infarction. (line 3).

  1. Regarding reperfusion injury, the infusion of hyperoxemic blood after infarct-related artery reperfusion could theoretically exacerbate the burst of reactive oxygen species, which are known to generate upon reperfusion of previously ischemic myocardium. If this were really what happens, myeloperoxidase levels would be higher with hyperoxemic reperfusion. However, an animal study (Spears et al. J Invasiv Cardiol 2002;14:160-166) demonstrated just the opposite, showing that this therapy actually reduce reactive oxygen radical damage. This is a reassuring finding, consistent with the observed beneficial effect of SSO2on infarct size, and should be mentioned in the manuscript.

Again, we thank the reviewer for pointing this out. We included this in a statement discussing the cautioning clinical issue on ambient oxygen supply as raised by AVOID and DETO2X and included the positive animal data as suggested by the reviewer (lines 135-140).

  1. The clinical experience of the authors with the SSO2therapy is only reported with a rough description (feasibility with no delay of primary PCI, no MACE during follow-up and a mention only regarding infarct size reduction). Indeed, no detailed data are presented (number of patients, patient characteristics, enrolling criteria and end-points, infarct location, imaging (CMR, rest perfusion SPECT9) data, periprocedural complications, infarct size reduction over time and clinical outcome), limiting the scientific value of this experience. Because the clinical experience with “optimized” SSO2 delivery (i.e. hyperoxemic blood infused through a 5Fr coronary diagnostic catheter into the ostium of the LM, infusion time reduced from 90 to 60 minutes increasing the flow rate from 75 mL/min to 100 mL/min) is limited (20 patients in the pilot study, 100 patients in the IC-HOT study), additional clinical data are needed to confirm the safety and salutary benefits of the “optimized” SSO2 delivery following primary PCI in patients with anterior STEMI.

A full-scale report including all the clinically relevant efficacy and safety outcomes is beyond the scope of the current “perspective” article. The numbers of patients treated is still too small to provide beyond a small case series. Therefore, we have deleted the reference to individual patient examples and mention the large outcome now requested by German authorities (lines 267-275).

Round 2

Reviewer 2 Report

I think that the revised manuscript has been sufficiently improved according to my suggestions.